# Raman Spectral Signatures of Serum-Derived Extracellular Vesicle-Enriched Isolates May Support the Diagnosis of CNS Tumors

**DOI:** 10.3390/cancers13061407

**Published:** 2021-03-19

**Authors:** Matyas Bukva, Gabriella Dobra, Juan Gomez-Perez, Krisztian Koos, Maria Harmati, Edina Gyukity-Sebestyen, Tamas Biro, Adrienn Jenei, Sandor Kormondi, Peter Horvath, Zoltan Konya, Almos Klekner, Krisztina Buzas

**Affiliations:** 1Laboratory of Microscopic Image Analysis and Machine Learning, Biological Research Centre, Institute of Biochemistry, Eötvös Loránd Research Network (ELKH), H-6726 Szeged, Hungary; bukva.matyas@brc.hu (M.B.); dobra.gabriella@brc.hu (G.D.); koos.krisztian@brc.hu (K.K.); harmati.maria@brc.hu (M.H.); sebestyen.edina@brc.hu (E.G.-S.); horvath.peter@brc.hu (P.H.); 2Department of Medical Genetics, Doctoral School of Interdisciplinary Medicine, University of Szeged, H-6720 Szeged, Hungary; 3Department of Applied and Environmental Chemistry, University of Szeged, H-6720 Szeged, Hungary; juan.gomez@chem.u-szeged.hu (J.G.-P.); konya@chem.u-szeged.hu (Z.K.); 4Department of Immunology, Faculty of Medicine, University of Debrecen, H-4032 Debrecen, Hungary; biro.tamas@med.unideb.hu; 5Monasterium Laboratory, D-48149 Münster, Germany; 6Clinical Centre, Department of Neurosurgery, University of Debrecen, H-4032 Debrecen, Hungary; jenei.adrienn@med.unideb.hu (A.J.); klekner.almos@med.unideb.hu (A.K.); 7Department of Traumatology, University of Szeged, H-6720 Szeged, Hungary; kormondi.sandor.pal@med.u-szeged.hu; 8Department of Immunology, University of Szeged, H-6720 Szeged, Hungary

**Keywords:** small extracellular vesicles, Raman spectroscopy, glioblastoma multiforme, brain metastasis, meningioma, CNS tumors, liquid biopsy

## Abstract

**Simple Summary:**

The conventional central nervous system (CNS) tumor diagnostic methods, especially the invasive intracranial surgical tissue sample collecting, imposes a heavy burden on both patients and healthcare providers. We aimed to explore the potential role of serum-derived small extracellular vesicles (sEVs) in diagnosing CNS tumors through Raman spectroscopic analyses. A relevant number of clinical samples (138) were obtained from four patient groups, namely glioblastoma multiforme, brain metastasis of non-small-cell lung cancer, meningioma, and lumbar disc herniation as controls. After the isolation and Raman measurements of sEV-sized particles, the Principal Component Analysis–Support Vector Machine algorithm was performed on the Raman spectra for pairwise classifications. The groups compared were distinguishable with 80–95% sensitivity and 80–90% specificity. Our results support that Raman spectroscopic analysis of sEV-sized particles is a promising liquid-biopsy-based method that could be further developed in order to be applicable in the diagnosis of CNS tumors.

**Abstract:**

Investigating the molecular composition of small extracellular vesicles (sEVs) for tumor diagnostic purposes is becoming increasingly popular, especially for diseases for which diagnosis is challenging, such as central nervous system (CNS) malignancies. Thorough examination of the molecular content of sEVs by Raman spectroscopy is a promising but hitherto barely explored approach for these tumor types. We attempt to reveal the potential role of serum-derived sEVs in diagnosing CNS tumors through Raman spectroscopic analyses using a relevant number of clinical samples. A total of 138 serum samples were obtained from four patient groups (glioblastoma multiforme, non-small-cell lung cancer brain metastasis, meningioma and lumbar disc herniation as control). After isolation, characterization and Raman spectroscopic assessment of sEVs, the Principal Component Analysis–Support Vector Machine (PCA–SVM) algorithm was performed on the Raman spectra for pairwise classifications. Classification accuracy (CA), sensitivity, specificity and the Area Under the Curve (AUC) value derived from Receiver Operating Characteristic (ROC) analyses were used to evaluate the performance of classification. The groups compared were distinguishable with 82.9–92.5% CA, 80–95% sensitivity and 80–90% specificity. AUC scores in the range of 0.82–0.9 suggest excellent and outstanding classification performance. Our results support that Raman spectroscopic analysis of sEV-enriched isolates from serum is a promising method that could be further developed in order to be applicable in the diagnosis of CNS tumors.

## 1. Introduction

In recent decades, secreted extracellular vesicles (EVs) have been recognized as a pathway for intercellular communication in both eukaryotes and prokaryotes [1]. Furthermore, several studies demonstrate the role of EVs in maintaining cellular homeostasis and integrity by compensating for the stress condition [2,3,4]. Their involvement in different pathophysiological processes has already been highlighted, especially in malignant diseases [5,6]. EVs released by tumor cells are involved in both stromal and distant cell communication, metastatic niche formation, and immune cell suppression [7,8,9,10,11,12,13].

Recent clinical research has highlighted that EVs could serve as novel tools for various therapeutic approaches, including oncotherapy, vaccination, immune-modulatory or regenerative therapies, and drug delivery [14]. Furthermore, EVs are gaining increasing popularity in biomarker research as their potential in liquid biopsy has been recognized [15].

Secreted EVs are stably present in body fluids, and represent a concentrated sample of the cytosolic milieu (proteins, nucleic acids, and lipids) of the donor cells [16,17,18]. It has been shown that EVs isolated from the serum and plasma offer a useful tool to improve the signal-to-noise ratio in analytics by assuring to abundant protein depletion (such as albumin and lipoproteins) and enriching the tumor-specific molecular composition [19,20]. Moreover, EVs can cross various biological barriers, such as the blood–brain barrier (BBB), and easily enter the peripheral blood [21,22].

Examining the protein, nucleic acid, or lipid contents of EVs has revealed several molecules as promising diagnostic markers for different tumor types. For example, glypican-1 glycoprotein enriched in circulating EVs has been shown to be suitable for distinguishing malignant pancreatic cancer from benign malformations with 100% classification accuracy [23,24].

Given their favorable biological properties, serum-derived EVs are being evaluated in the diagnosis and monitoring of central nervous system (CNS) tumors which represent a major challenge in oncology [25].

Today, the diagnosis of CNS tumors mainly relies on neuroimaging techniques (e.g., magnetic resonance imaging (MRI) or computer tomography (CT)) and tissue biopsy. However, all of these methods have numerous limitations [26]. Among others, MRI can only detect tumor masses of sufficient magnitude, and has little prognostic value in terms of long-term recurrence [27]. Distinguishing between different CNS malignancies, such as glioblastoma multiforme and brain metastases, is also challenging using neuroimaging techniques [28]. In addition, treatment-related changes can overlap with residual or recurrent tumors, making tumor monitoring highly challenging [29].

Many brain tumors are particularly difficult to be sampled or are inaccessible for tissue biopsy. Even in cases of biopsy, the procedure harbors significant risks for the patient (e.g., hemorrhage, or impairment of neurological functions). These risks and difficulties hamper not only the diagnosis, but also the monitoring of treatment response or distinguishing tumor recurrence from pseudoprogression [30]. In addition, in some cases, such as in glioblastoma multiforme, the focal sampling of small and localized tumor tissues may not fully capture intratumoral heterogeneity [31,32].

Liquid biopsy has remarkable advantages over conventional methods, offering a minimally invasive, safer, faster, and cheaper way to diagnose and monitor malignant diseases. Tumor tissues release various types of biomarkers, such as proteins, nucleic acids or lipids, and EVs that accumulate in body fluids (including the blood, urine, cerebrospinal fluid and saliva) are accessible for sampling [33,34,35].

Tumor markers determined from blood samples, such as the prostate-specific antigen (PSA), alpha-fetoprotein and cancer antigen 125 (CA-125), have already been introduced into clinical practice to support the diagnosis and/or monitoring of prostate, liver and ovarian cancers. Research is underway to identify other non-invasive biomarkers for the monitoring of a broader range of malignant diseases [36].

However, identifying blood-based CNS tumor markers is more challenging, presumably explained by several reasons. BBB can prevent tumor-derived molecules (tumor “information”) from entering the peripheral blood, therefore molecules released by other tissues/cells at high concentrations can impede the detection of potential tumor biomarkers present in lower concentrations. Abundant serum proteins (such as albumin or lipoproteins) also appear as a significant analytical noise [19,37]. Nevertheless, investigations for blood-based CNS tumor markers are in the spotlight of neuro-oncological research, as they would have outstanding advantages in patient care [38].

Due to their beneficial properties detailed above, EVs are promising tools in the research for CNS tumor biomarkers. Several published studies aimed to examine the nucleic acid and protein contents of blood samples or EVs derived from CNS tumor patients (specifically, from patients with gliomas). However, these studies attempted to identify one or two biomarkers of nucleic acid or protein types, and these molecules did not prove to be sufficiently specific or sensitive to serve as diagnostic markers, and thus they were not validated with blinded samples [39,40,41].

Analyzing the whole molecular composition of tumor-related EVs isolated from blood samples may provide a solution to overcome the difficulties encountered in CNS tumor biomarker research. Raman spectroscopy is a suitable approach for this purpose, since it provides information on the entire molecular content of a sample. Raman spectroscopy is a non-destructive, label-free vibrational technique that measures the non-elastic scattering effect induced by a radiating laser. The energy of this inelastically scattered light is reduced by the vibrational energy of the chemical bonds present in the molecules within a sample [42]. The difference is proportional to, and thus specifically refers to, the chemical composition of the sample. Therefore Raman spectroscopy can reveal a specific spectral signature that describes the whole chemical composition, and thus avoids the need for identifying any specific protein, nucleic acid, or lipid biomarkers [43]. In addition, Raman spectroscopy may be suitable for the characterization of EVs by identifying different subtypes by origin and function, which is an important and long-standing challenge for EV-based biomarker research [44].

Recent studies suggest that Raman spectroscopic analysis of the whole molecular composition of various sample types may be suitable to develop promising diagnostic methods for clinical practice [45,46,47]. Some of these in vitro studies focused on EVs and demonstrated their outstanding diagnostic efficiency. For example, using this technique, Parks and colleagues distinguished EVs released by lung cancer cells from those secreted by normal cells with 95.3% sensitivity and 97.3% specificity. Meanwhile, discriminatory spectral differences were also identified using principal component analysis (PCA) [48]. Charmichael and colleagues revealed that EVs originating from pancreatic cancer cells were distinguishable from those released by normal pancreatic epithelial cells with 90% accuracy [49].

However, no studies to date have investigated the diagnostic efficiency of Raman spectroscopic analysis of serum-derived EVs with regard to CNS tumors. Thus, we aimed to explore the potential role of serum-derived EVs in diagnosing CNS tumors through Raman spectroscopic analyses on a relevant number of clinical samples. 

For this purpose, 138 serum samples were obtained from four patient groups. The serum samples were collected from patients diagnosed with the two most common types of brain tumors, namely malignant glioblastoma multiforme (GBM) and the typically benign meningioma (M), as well as from patients with a prevalent brain metastasis originating from non-small-cell lung cancer (BM). Patients with lumbar disc herniation without evidence of neurological cancer served as controls (CTRL) [20,50,51]. Particles within the size range of small EVs (sEVs) were isolated from serum samples via differential centrifugation, and were assessed by a Raman microscope. Multivariate analyses, including Principal Component Analysis–Support Vector Machine (PCA–SVM) and FreeViz, as well as conventional statistical methods, such as Receiver Operating Characteristic (ROC) analysis, were carried out on spectroscopic data to develop and evaluate a classification model.

Our results support that analyzing the serum-derived sEV-enriched isolates by Raman spectroscopy, which captures the whole molecular composition, may be suitable to develop a method with a possible diagnostic value for CNS tumors, and thus it may have the potential to be introduced into clinical practice in the future.

## 2. Results

### 2.1. Particles Isolated from Serum Show sEV Properties

Particles were isolated by differential centrifugation from 138 serum samples of patients with GBM, BM), M and CTRL. Isolated sEVs were characterized by transmission electron microscopy (TEM) and nanoparticle tracking analysis (NTA), as well as by examining characteristic sEV markers (Alix, CD81 and calnexin) by Western blotting (WB) (Figure 1). Average concentration, mean and mode diameter of the particles were measured as 7.41 × 10^10^ particles/mL, 111.20 nm and 83.32 nm, respectively. Alix, CD81 positivity and calnexin negativity was determined (see Appendix A for the original WB images).

No statistically significant differences were identified among the patient groups in any of the parameters of the isolated particles.

### 2.2. Patient Groups Can Be Distinguished Using the PCA–SVM Algorithm with High Classification Efficiency

Raman spectroscopic analyses of the isolated 138 samples yielded 5 spectra per sample. The spectral range between 801 cm^−1^ and 3100.5 cm^−1^ was investigated. After standard normal variate (SNV) normalization and PCA transformation, the classification of samples was performed using the SVM algorithm. Classification efficiency was evaluated by classification accuracy (CA), sensitivity, specificity and the area under the curve (AUC) value derived from the ROC analysis. Relevant spectral differences were revealed by PCA. Figure 2 shows the flowchart of Raman spectroscopy data processing.

After averaging the spectra, row normalization was performed using the SNV method (Step 1) (Figure 3). 

Following SNV-normalization, the spectra for the samples of the four patient groups were compared pairwise (each patient group was compared to the control, and BM vs. GBM was compared) for two purposes: first, to develop and test a classification algorithm, and second, to identify relevant spectral differences. PCA applied on the pairwise comparisons reduced multivariate data dimensions by transforming the original variables (wavenumbers) into a smaller number of new variables, i.e., principal components (PCs) (Step 3). 

Pairwise comparisons were conducted using the linear SVM (Step A4) algorithm, yielding classification models for each paired group. To make predictions for the test samples, a minimum threshold for the group-membership score was determined. Test samples with scores above this threshold were classified into the target group of interest. The optimal score thresholds were automatically set to correspond to the highest classification accuracy (CA, the ratio of correctly classified samples per all samples).

CA was 85.6% for CTRL vs. GBM, 91.4% for CTRL vs. BM, 82.9% for CTRL vs. M and 92.5% for BM vs. GBM. The best classification performance was achieved when a certain number of PCs were included in the models: 30 PCs for CTRL vs. GBM, 38 PCs for CTRL vs. BM, 27 PCs for CTRL vs. M, and 26 PCs for BM vs. GBM (Figure 4).

Sensitivity and specificity were evaluated as further metrics of classification performance. ROC analyses of the pairwise classification models yielded four graphs showing the automatically set optimal thresholds (having the highest CA value), with related sensitivity, specificity and AUC values, as well as *p*-value (Figure 5).

As shown in the graphs in Figure 5, using the optimal thresholds, the classification models were able to distinguish GBM, BM and M patients from CTRL patients with a sensitivity and specificity of 90% and 80%, 93.75% and 90%, 80% and 85%, respectively (Step A5). Using the classification model, the two malignancies, BM and GBM, could be distinguished from each other with a sensitivity of 98% and a specificity of 83.3%. In the same order of pairwise comparisons (GBM, BM and M patients vs. CTRL, and BM vs. GBM), the AUC values were 0.87, 0.95, 0.82 and 0.9, respectively (*p* < 0.0001 in all cases).

### 2.3. Analysis of the PCs Revealed Discriminative Spectral Differences

Next, differences in the molecular content of serum-derived sEV-enriched isolates from each group were investigated to reveal the spectral differences relevant with regard to the classification. SNV-normalized spectra and the PCs obtained from PCA were analyzed using the FreeViz method, in order to reveal and visualize relevant spectral differences. 

The FreeViz method (Step B4) displayed the optimized projections of the multivariate data sets in a 2-dimensional scatterplot (Figure 6). Based on the length and direction of PC vectors, two PCs that were revealed to play the most important role in distinguishing each paired group (marked with a yellow background in Figure 6) were further assessed to determine discriminative spectral signatures.

Based on the results of the FreeViz method and p-values from Welch’s *t*-test, PC14 and PC2, PC9 and PC13, P10 and PC19, and PC2 and PC3 explained most of the discriminative differences in the CTRL vs. GBM, CTRL vs. BM, CTRL vs. M and BM vs. GBM comparisons, respectively (*p* < 0.05 in all cases) (see Appendix A for the score plots of the selected PCs).

Evaluating the selected PCs, we attempted to find the chemical bonds and functional groups corresponding to the spectral differences found to have an important role in distinguishing the compared groups (Step B5).

Regarding the CTRL vs. GBM comparison, most of the discriminative spectral differences were characteristic for carbohydrates, such as bands associated with a pyranose ring (800–975 cm^−1^), O-H deformation vibrations (1030–1080 cm^−1^) and C-O stretching vibrations (1030–1290 cm^−1^). These bands largely overlap with the region’s characteristic for nucleic acids, including the bands associated with the vibrations of the phosphate-sugar backbone (800–1000 cm^−1^), symmetric and asymmetric phosphate group stretching vibrations (1000–1250 cm^−1^), glycosidic bond vibrations (1250–1550 cm^−1^), and in-plane double bond vibrations of bases (1530–1780 cm^−1^) (Figure 7).

Regarding the CTRL vs. BM comparison, the wavenumbers found to have an important role in distinguishing the BM group from the control mainly correlated with lipids (CH_3_ asymmetrical bending (1470–1490 cm^−1^), CH_2_ and CH_3_ symmetrical and asymmetrical stretching vibrations (2700–3100 cm^−1^)) and amino acids (–NH_3_^+^ deformation band (1485–1150 cm^−1^), –NH_3_^+^ asymmetrical stretching (3000–3100 cm^−1^), carboxylate ion stretching (1560–1600 cm^−1^) and C=O stretching vibrations of the carboxyl group (1700–1755 cm^−1^)). Regarding the CTRL vs. M and BM vs. GBM comparisons, the wavenumbers highly correlated with vibrations originating from acyl chains of lipids, such as CH_3_ and CH_2_ symmetric and asymmetric stretching vibrations (2700–3100 cm^−1^) (see Appendix A for the tabular form of the discriminative spectral differences).

## 3. Discussion

Circulating sEVs are considered as promising sources of CNS tumor markers. Several studies have investigated the nucleic acid and protein contents of blood samples or EVs from CNS tumor patients. These studies have generally attempted to identify one or two biomarkers targeting the proteome, genome or lipidome. However, these molecules alone do not have sufficient diagnostic or prognostic value, thus they cannot be used as single biomarkers, and none of these have been validated on blinded clinical samples [26,39,40,41,52,53].

Analyzing the entire molecular composition of tumor-related EVs could provide a solution to overcome the difficulties encountered in CNS tumor biomarker research. Raman spectroscopy is a suitable approach for this purpose, as it provides information on the total molecular content, yielding a specific spectral signature that describes the chemical composition of a sample. Thus, it has the potential to avoid the need for identifying any specific protein, nucleic acid or lipid biomarkers [43].

Based on these considerations, we have attempted to explore the potential role of serum-derived sEVs in the diagnosis of CNS tumors through Raman spectroscopic analyses on a clinically relevant cohort. According to our knowledge, this is the first study that aims to classify CNS tumors based on the Raman spectra of sEV-enriched isolates from serum samples.

For this purpose, 138 serum samples obtained from four patient groups were analyzed. Serum samples were collected from three brain tumor groups considered as the most common malignant, benign and metastatic brain tumors (GBM, BM, M) and from a control group (CTRL) [20,50,51]. sEV-sized particles from serum samples were isolated by differential centrifugation.

The particles found in the isolates show sEV properties (CD81, Alix positivity and calnexin negativity). However, since abundant serum protein aggregates and lipoproteins (LPs) are able to mimic sEVs in terms of size (mean and mode diameter of 111.20 nm and 83.32 nm), we cannot state that only sEVs are present in the isolates. In our previously published proteomic-based study on the same patient groups, we have shown that, although contaminants are still present in the isolates, differential centrifugation significantly enriched the sEV-specific markers and reduced the level of LPs. Since LPs and abundant serum protein aggregates are certainly present in addition to sEVs, the isolates should be considered only as sEV-enriched rather than purified sEVs. In light of these, Raman spectra may characterize a circulating particle profile, part of which is sEVs assumed as biomarkers.

No significant differences were found between the four patient groups in the concentration, mean and mode diameter of sEV-sized particles. Osti et al. observed higher EV concentration in the plasma samples of GBM patients, brain metastases and extra-axial brain tumors compared to healthy controls [54]. Other researchers also showed higher EV concentration in tumor patients when unfractionated EV isolates or a broader spectrum of EVs were analyzed [55,56,57]. However, other non-neoplastic diseases of the central nervous system can also increase the number of sEVs, as has been shown in acute ischemic stroke or multiple sclerosis patients [58,59]. These findings suggest that the elevated sEV concentration cannot be clearly attributed to the presence of the tumor as immune responses or other systemic responses also contribute to the circulating EV population. Therefore, the intense inflammation associated with lumbar disc herniation (CTRL) may explain why no statistical difference was identified between tumorous and non-tumorous patient groups [60].

In the light of the above, we hypothesize that the isolates contain not only tumor tissue-derived vesicles but also other circulating vesicles, including vesicles released by red blood cells, platelets and immune cells. Therefore, the differences observed in the Raman spectra of the different patient groups may not only reflect tumor-specific processes but other host responses, i.e., the tumor-associated immune responses or different coagulant phenotypes as well [61,62,63].

After the Raman spectroscopic measurements, multivariate analyses and conventional statistical methods were applied on the spectroscopic data to develop and evaluate a classification model, and find the characteristic spectral signatures distinguishing between the patient groups and healthy controls, as well as between the glioblastoma multiforme and brain metastasis groups.

PCA was applied to all the SNV-normalized spectra. PCA is a standard way to reduce data dimensionality and obtain characteristic spectral signatures [48,64].

Classification was performed by applying the SVM algorithm on PCA-transformed data. Classification performance was evaluated by CA, sensitivity (rate of true positive samples), specificity (rate of true negative samples), and the AUC value derived from ROC analysis, which are all commonly used and accepted metrics in clinical practice.

The GBM, BM and M groups proved to be distinguishable from CTRL with 85.6%, 91.4%, 82.9% of CA, respectively. Interestingly, maximal classification accuracy depended on the number of PCs used for classification, showing an increasing trend towards a specific number of PCs (Figure 4). The relationship between the number of PCs included and CA achieved is probably explained by the complexity of these biological samples.

In most studies, the first two PCs (PC1 and PC2) were able to describe the complete data set and revealed distinctive patterns [64]. However, as Lyng and colleagues’ findings show, the first two components may not sufficiently explain the information included in the complete Raman spectrum for biological samples, due to their complex molecular composition [65]. Using combinations of PCA and various discriminant analyses, Lyng and colleagues found that 20 PCs were required to separate breast tumor tissue samples from healthy controls with 80% CA. Our results also support that including two PCs, only one cannot develop an accurate classification model capable of spectrally discriminating between different ex vivo biological sample groups. However, classification performance can be improved by increasing the number of PCs included in the model, although above a certain number of PCs used, the information they explain may be meaningless or may account for noise, leading to decreased classification accuracy (Figure 4). This suggests that the spectra for biological samples show a high degree of overlap due to their complexity. Hence, accurate classification can be performed only when one correctly uses several dimensions, taking small spectral differences into account.

Although sensitivity and specificity can be calculated by regarding each value of the group-membership scores as a threshold, the ROC curves, including all possible decision thresholds, plus AUC together, offer a more comprehensive assessment [66,67].

According to the ROC analyses, sensitivity and specificity values were as follows: 90% and 80% for CTRL vs. GBM, 93.75 and 90% for CTRL vs. BM, 80% and 85% for CTRL vs. M, and 98% and 83.3% for BM vs. GBM, respectively. In the same order of comparisons, AUC values were 0.87, 0.95, 0.82 and 0.9 (Figure 5).

Based on the literature of ROC analysis, our classification models for CTRL vs. GBM and CTRL vs. M comparisons can be considered as “excellent”, and “outstanding” for CTRL vs. BM and BM vs. GBM comparisons [66].

Due to its reliable theoretical basis, SVM has become one of the most widely used classification methods in recent years, especially for complex multivariate data sets obtained from spectroscopic analyses, characterized by high variance and probable outliers [65,68,69,70]. These properties make the SVM classifier particularly suitable to discriminate between clinical samples based on their Raman spectra, even for diseases known to be highly heterogeneous (such as GBM) [31,32]. Furthermore, Neska-Matuszewska highlighted that various malignancies (e.g., BM and GBM) are challenging to be distinguished using conventional neuroimaging techniques [28]. In light of our findings, Raman spectra-based SVM classification may support a reliable differential diagnosis between primary brain tumors and metastatic brain malignancies. However, it should be noted that the future confirmation of our results via the comparison of other primary and metastatic brain tumor types is clearly required.

Using the FreeViz method, PCs that were particularly important in terms of distinguishing between the compared groups could be identified (Figure 6). By examining the contribution of the wavenumbers to the selected PCs, we attempted to find the chemical bonds and functional groups that correlate with the spectral differences revealed to play an important role in our pairwise classifications. 

The molecular correlation of the vibrational bands in the Raman spectra is extremely difficult to interpret. The difficulty arises from the complexity of biological samples in which an abundance of organic molecules coexist and share some of the functional groups responsible for the Raman-spectral features [71]. As a result, the overlap of different vibrational bands hinders the precise identification of any specific molecules based on Raman spectral features (Figure 7). Nevertheless, it was possible to identify discriminative spectral differences in the CTRL vs. GBM comparison, defining bands characteristic for carbohydrates and nucleic acids. These differences may be due to the characteristic metabolism of GBM, as it is associated with a significant increase in glycolysis for energy production and abnormal purine and pyrimidine synthesis [72]. Comparing the spectra of the CTRL and BM groups, significant differences were found in the characteristic bands of lipids and amino acids, which can be partly explained by the fact that an NSCLC appears to be reliant on fatty acid and serine catabolism [73]. Comparing the CTRL and M groups, as well as the two malignant groups BM and GBM, the lipid bands had outstanding importance with regard to discriminatory differences. The prominent importance of lipids in the BM vs. GBM comparison may be explained by the increased lipid catabolism of NSCLC and the elevated level of de novo lipid synthesis in GBM [72,73]. However, more detailed identification based on the difficulties described above is not expedient for complex biological samples.

It should be noted that co-purification of abundant serum proteins and LP particles in EV isolation methods is a common and well-known challenge [74]. Liu and colleagues emphasized that serum is not the perfect choice for representative sampling of circulating EVs, as a high proportion of EVs may be lost during clotting, and blood components enrolled in the coagulation may also (e.g., platelets) release EVs altering the original content of blood samples [58]. Some cancerous diseases, such as GBM, may also have a procoagulant phenotype [75].

Despite these difficulties, we have revealed in a previously published article that EV isolation from the serum samples of the same patient groups significantly improves the signal-to-noise ratio, even in the case of GBM with an elevated procoagulant activity [20]. Although abundant serum proteins and LPs were still present in EV isolates, isolation depleted their concentration and enriched the EV and tumor-specific protein markers. These results are consistent with previous similar researches on serum-derived EVs [19,76]. Nevertheless, examining plasma instead of serum should be considered in further investigations [58,74].

Enciso-Martinez and colleagues have determined Raman spectral signatures which were able to distinguish EVs from LPs and platelets with 95% confidence [77]. These special signature regions were found at 1004 cm^−1^ and between 2811 cm^−1^ and 3023 cm^−1^. Wavelength 1004 cm^−1^ had a strong peak in EVs but was not present in LPs and platelets. Furthermore, in the spectral range of 2811–3023 cm^−1^, EVs showed stronger intensity after 2900 cm^−1^ (‘protein component of the CH region’) compared to the spectra of LPs, where the region before 2900 cm^−1^ (’lipid component of the CH region’) proved to be more intense.

The Raman spectra of sEV-enriched isolates in our study show similar properties: a peak with strong intensity is present at 1004 cm^−1^ and the “protein component” of the CH region was found to be much more prominent than the “lipid component”.

Considering its feasibility and beneficial properties, the steps of our research work could be incorporated into method developments aiming to establish novel diagnostic tools potentially applicable in clinical practice. Our isolation protocol has several advantages, as it does not require expensive equipment or highly trained professionals, and the entire procedure (along with characterization) is performed in about 4 hours. 

Although some isolation methods, such as size exclusion chromatography and precipitation, can be performed more quickly, isolation via differential centrifugation results in fewer particles in the size range potentially LPs, lower intensity of LP markers, higher 61–150 nm EVs to 0–60 nm EVs ratio, and higher intensity of EV markers [78]. Raman spectroscopy provides a comprehensive analysis of the circulating tumor-related molecular content. Besides, Raman spectroscopy has additional advantages, such as operator safety, elimination of disposables and analysis waste, fast analytical response of less than 2 min, reduction of the risk of errors because no intrusion or dilution are needed, and negligible maintenance costs. By employing the appropriate preprocessing steps, classification requires reduced computational time and capacity. Moreover, SVM classification based on Raman spectra is suitable to support the proper assessment of even complex biological samples, despite their high degree of variance. This approach may also support decision-making in challenging clinical cases, such as distinguishing between primary brain tumors and other metastatic brain malignancies.

Besides its advantages, our approach also has limiting factors. LPs and protein aggregates in the same size range of sEVs may co-isolate during the differential centrifugation. Accordingly, it is recommended to refer to the isolates as “particle profile with sEVs”, “sEV-sized particles”, or “sEV-enriched isolates”. Because of this heterogeneity, it is also not evident whether the Raman-based classification differentiates the tumor-specific molecular information concentrated in the circulating sEVs or different type of particles. Isolation purity could be improved by combining different isolation methods and examining plasma instead of serum [78].

Isolates from serum may be enriched not only with tumor tissue-derived sEVs but also with EVs released by red blood cells, platelets and immune cells. As it is not revealed whether sEVs from other sources are analytical noise or carriers of relevant information, it might be worthwhile to distinguish tumor tissue-derived sEVs based on surface markers and to perform Raman spectroscopic analyses only on them in the future [76].

In conclusion, our results provide a proof of principle for a novel detection technology that might be utilized to develop a relatively easy-to-execute and appropriate method, which could have the potential to support and simplify the diagnosis and monitoring of CNS tumors in the future. However, clinical applicability definitely requires further development.

## 4. Materials and Methods

### 4.1. Patients

Blood samples of 138 patients treated at the Department of Neurosurgery at the University of Debrecen were analyzed. Samples were obtained from patients with GBM, BM, M. Patients with spinal disc herniation (a non-cancerous CNS disease) served as control CTRL (Table 1).

Each patient signed an informed consent form. The study was conducted in accordance with the Declaration of Helsinki, and ethical approval was obtained from two independent bodies (51450-2/2015/EKU (0411/15), Medical Research Council, Scientific and Research Ethics Committee, Budapest, October 30, 2015 and 121/2019-SZTE, University of Szeged, Human Investigation Review Board, Albert Szent-Györgyi Clinical Centre, Szeged, 19 July 2019)

### 4.2. Preparation of Serum Samples, sEV Isolation and Characterization

Preparation of serum samples was described in our previously published article [20].

Briefly, after 1 h of blood clotting at room temperature, sEV isolation from serum samples was performed via differential centrifugation (20 min at 3000× *g*, 10 °C; 30 min at 10,000× *g*, 4 °C; 70 min at 100,000× *g*, 4 °C). After the last centrifugation step, the pellet was resuspended in Dulbecco’s phosphate-buffered saline (DPBS) and was stored at −80 °C until further processing. 

To characterize sEVs, we followed the main suggestions and requirements included in the guideline ‘Minimal Information for Studies of Extracellular Vesicles 2018’ (MISEV 2018) [17].

sEVs were diluted in particle-free DPBS and analyzed using a NanoSight NS300 instrument with 532 nm laser (Malvern Panalytical Ltd., Malvern, UK). Six videos of 60 s were recorded for each sample under constant settings (Camera level: 15; Threshold: 4, 25 °C; 60–80 particles/frame) and analyzed to obtain data on size distribution and particle concentration.

Classical EV markers were presented by Western blot analyses using NuPAGE reagents and an XCell SureLock Mini-Cell System (Thermo Fisher Scientific, Waltham, MA, USA) according to the manufacturer’s protocols. For detection of the CD81, Alix and Calnexin markers, we used rabbit anti-human CD81 (1:1000, Sigma-Aldrich, St. Louis, MO, USA), rabbit anti-human Alix (1:1000, Sigma-Aldrich, St. Louis, MO, USA) and rabbit anti-human Calnexin (1:10,000), Sigma-Aldrich, St. Louis, MO, USA) primary antibody and HRP-conjugated anti-rabbit IgG (1:1000, R&D Systems, Minneapolis, MN, USA) secondary antibody. THP-1 cell line (ATCC, Teddington, UK) lysate was used for positive control for Calnexin.

In order to examine sEV morphology, TEM analysis was performed using a Tecnai G2 20 X-Twin type instrument (FEI, Hillsboro, OR, USA), operating at an acceleration voltage of 200 kV. For TEM measurements, the samples were dropped on a grid (carbon film with 200 Mesh copper grids (CF200-Cu, Electron Microscopy Sciences, Hatfield, PA, USA) and dried without staining or other fixation procedure.

### 4.3. Raman Spectroscopy

Raman characterization of sEVs was carried out with a Senterra II microscope (Bruker) in backscattering configuration. The samples were centrifuged, drop-casted on a calcium fluoride substrate and air-dried at room temperature before the analysis. All the samples were analyzed using the same configuration parameters based on preliminary studies: nominal laser power 12.5 mW, integration time 30 s (2 coadditions), interferometer resolution 1.5 cm^−1^, excitation wavelength 532 nm. The spectra from all the samples were collected by using a 50× optical objective (Olympus). The described optical setup produces a laser spot of approx. 15 µm, which is the sampling area of the Raman spectra, and it is much smaller than the average size of the air-dried sample of approx. 4 mm, thus the Raman microscope operator can finely tune the position of the sampling spot and avoid duplication. The spectra were baseline-corrected before being averaged (5 spectra per sample) using the OPUS software available with the Bruker equipment. Spectral range between 801 cm^−1^ and 3100.5 cm^−1^ was used for further analyses (Appendix A).

### 4.4. Data Adjustment 

Row normalization of baseline-corrected data was performed using the SNV method. SNV transformed the mean to 0 and standard deviation to 1, making all spectra comparable in terms of intensity.

PCA with unit variance scaling was applied on the SNV-normalized spectra [79]. PCA served to reduce the dimensions of multivariate data by transforming the original variables (wavenumbers) into a smaller number of new variables, i.e., the PCs. 

Data adjustment was performed using the Orange 3.27.0 software (Ljubljana, Slovenia).

### 4.5. Classification

To develop and test a classification algorithm, the spectra for the samples from the four patient groups were compared pairwise (each patient group was compared to the control, and BM vs. GBM was compared). Sample classification was carried out using the linear SVM algorithm, yielding classification models for each paired group. First, the data were randomly split into train and test sets in a ratio of 90:10. Using the train set, SVM attempts were executed to find a hyperplane that can separate the compared groups in the PCA-transformed space. The process yielded a trained SVM model. Then, the trained SVM model ordered group-membership scores (from 0 to 1) to the test samples based on their positions and distances from the separating hyperplane. In practice, the decisions were made based on the location of the test samples from the plane, which is expressed by their group-membership scores. To make predictions about the test samples, a minimum threshold for the group-membership score was determined. Test samples with scores above this threshold were classified into the target group of interest. In each case, the train–test split was repeated ten times. 

Classification efficacy was assessed by sensitivity (proportion of correctly identified positive samples), specificity (proportion of correctly identified negative samples), and by the AUC value obtained from the ROC analysis [66]. Classification and efficacy evaluations were performed using the Orange 3.27.0 and GraphPad Prism 8.4.3 (San Diego, CA, USA) software packages.

### 4.6. Determining the Spectral Differences

The correlation between the obtained PCs and the different groups was determined by the FreeViz method [80]. Briefly, the FreeViz method displays multivariate data in a 2-dimensional scatterplot to separate samples from different patient groups. In the FreeViz plots, the samples and PCs are represented with dots and vectors, respectively (Figure 4). Since FreeViz optimized the display concerning the patient groups, the PCs that played a more important role in classification generally had longer vectors. Directions of the PC vectors were also revealing. When a region in the graph was mainly populated by samples of a certain group, the PC vectors in that direction could be regarded as good indicators of this group membership. The more a PC vector was approaching perpendicularity relative to the line separating the groups, the more useful it was for distinguishing them. Between-group statistical differences in PCs were analyzed using Welch’s *t*-test. Regarding that PCs are the linear combination of the original variables (wavenumbers), it is possible to determine the wavenumbers that have the largest contribution to a given PC. 

Values of *p* < 0.05 were considered significant. FreeViz was performed using the Orange 3.78.0 software [81].

## 5. Conclusions

Our study aimed to classify serum-derived sEVs from four patient groups based on their Raman spectral signatures. To the best of our knowledge, we are the first group to investigate the potential role of serum-derived sEVs in the diagnosis of CNS tumors using Raman spectroscopy. Based on various metrics, the classification efficiency proved to be excellent. In conclusion, our results support that Raman spectroscopic analysis of circulating sEV-enriched isolates is a promising liquid-biopsy-based method that could be further developed in order to be applicable in the diagnosis of CNS tumors. Our easy-to-perform analysis offers a novel detection technology that might be utilized in method developments aiming to simplify the diagnosis and monitoring of CNS tumors, and thus it might have the potential to be integrated into clinical practice in the future.

## Figures and Tables

**Figure 1 cancers-13-01407-f001:**
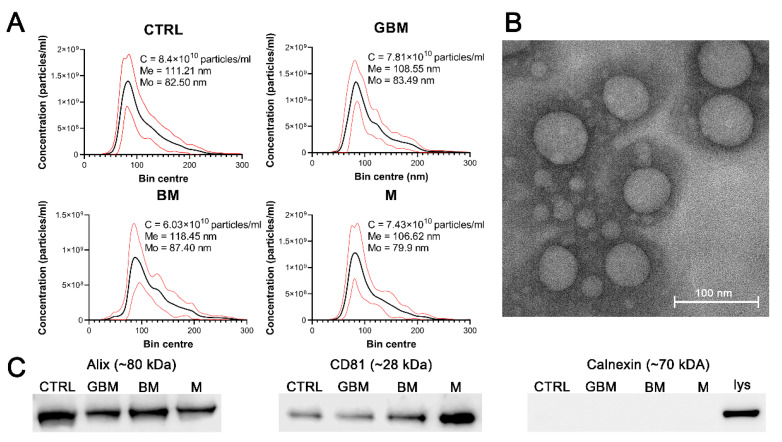
Characterization of the particles. The figure represents the results of the particle characterization: size distribution of the sEV samples isolated from the four patient groups (black and red lines represent the mean and the standard deviation of the concentration, respectively) (**A**), a representative TEM image of the sEVs (**B**), and the Western blot analysis of the sEV markers (**C**). (Abbreviations: CTRL, control; GBM, glioblastoma multiforme; BM, brain metastasis; M, meningioma; C, particle concentration; lys, cell lysate; Me, mean diameter size; Mo, mode diameter size.)

**Figure 2 cancers-13-01407-f002:**
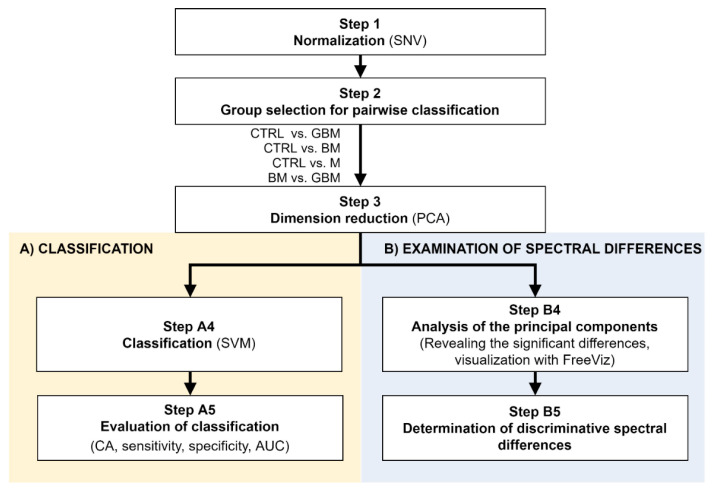
Workflow of Raman spectroscopic data processing. The figure shows the analysis step by step. After Step 3, the workflow separates (parts **A** and **B**) according to the purpose of the analysis. (Abbreviations: AUC, area under the curve; CA, classification accuracy; SNV, standard normal variate; SVM, support-vector machine; PCA, principal component analysis).

**Figure 3 cancers-13-01407-f003:**
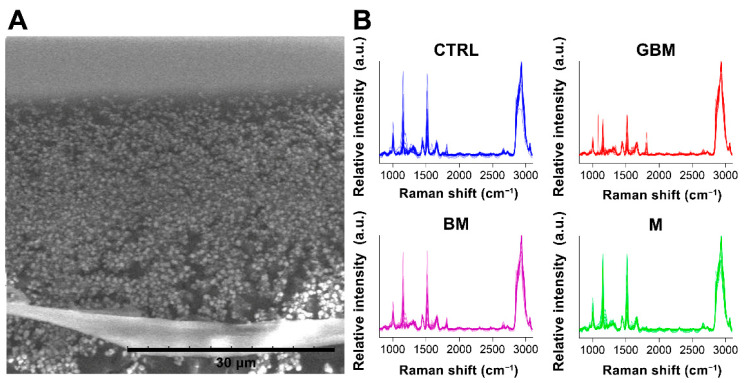
Particles on a calcium fluoride substrate (scanning electron microscope image) (**A**). Averaged and SNV-normalized spectra of the four patient groups (**B**). (Abbreviations: a.u., arbitrary unit).

**Figure 4 cancers-13-01407-f004:**
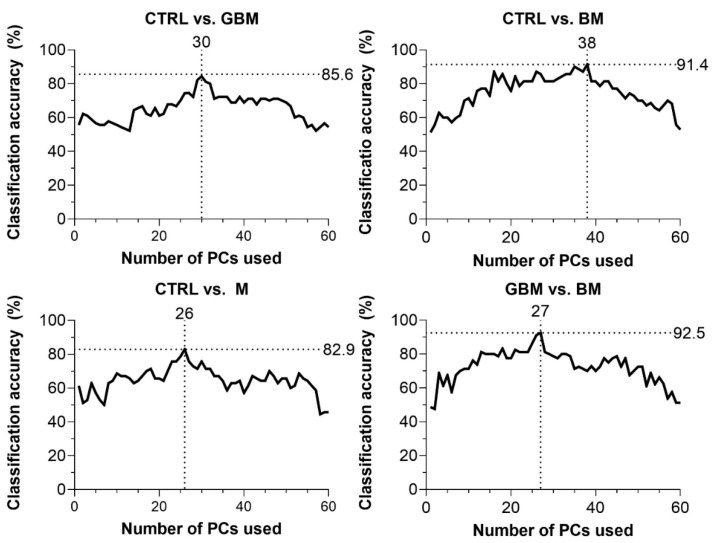
Classification accuracy (CA) scores with respect to the number of PCs included in the model (60 PCs at a maximum). Black dotted lines show the highest CA peaks with the corresponding number of PCs. (Abbreviations: PC, principal component).

**Figure 5 cancers-13-01407-f005:**
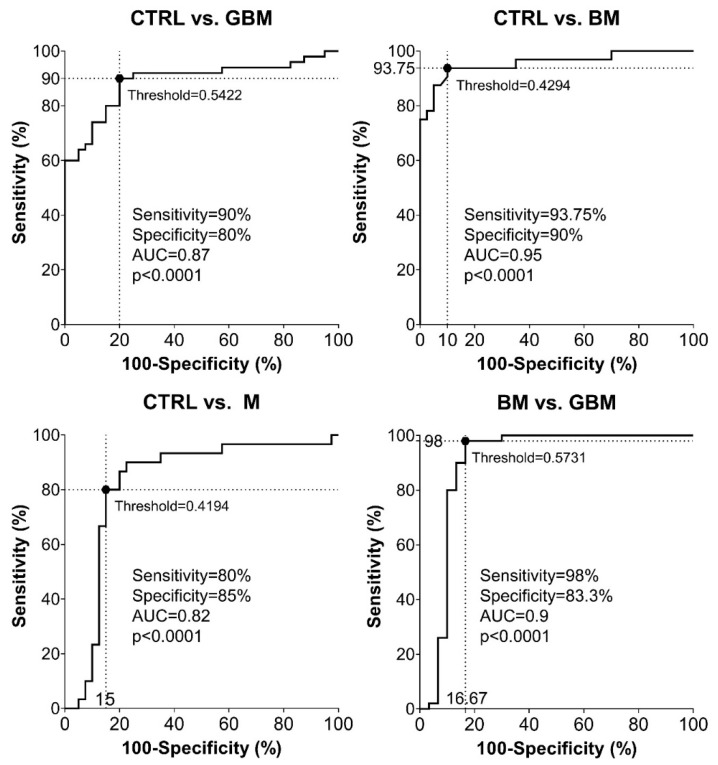
Receiver Operating Characteristic (ROC) curves for the classification models. Intersecting black dotted lines show sensitivity, specificity and corresponding threshold values of the group-membership score, with black filled circles at their intersections.

**Figure 6 cancers-13-01407-f006:**
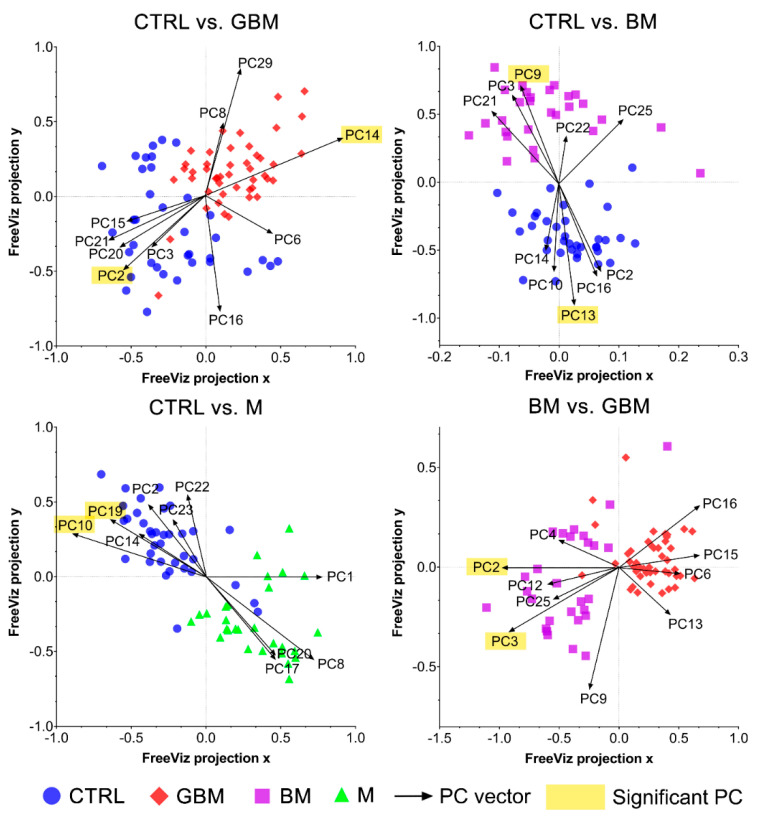
FreeViz projections of pairwise comparisons. Analysis of the PCA-transformed data using the FreeViz method yielded four graphs. Different dots and colors represent the patient groups and healthy controls. Black vectors represent the PCs. In each graph, only the 10 most relevant PC vectors were plotted. For each comparison, PCs marked with a yellow background indicate the 2 most significant PCs.

**Figure 7 cancers-13-01407-f007:**
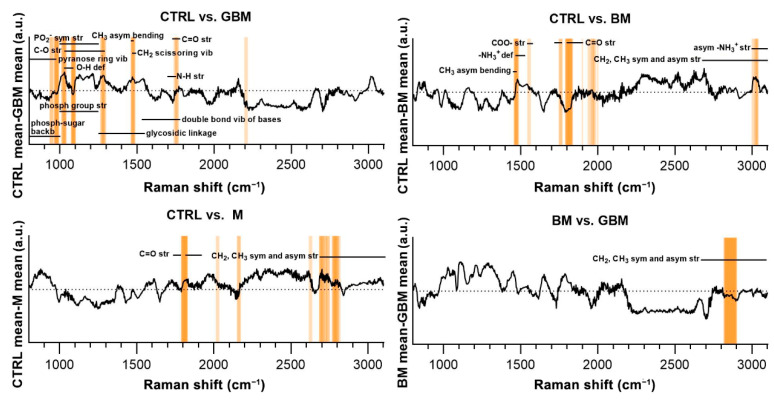
Subtraction spectra for the pairwise comparisons. Subtraction spectra were produced by subtracting the mean signal intensities for the groups compared. Spectral regions having a higher-than-average contribution to significant PCs were marked with orange bars. The more saturated a bar is, the more that region is represented on the selected PCs. The dotted horizontal line represents zero difference at y = 0. (Abbreviations: asym, asymmetric; backb, backbone; def, deformation; phosph, phosphate; str, stretching; sym, symmetric; vib, vibration).

**Table 1 cancers-13-01407-t001:** Patient cohort.

Patient Groups	No. of Patients	Age (years)			Sex	
		Range	Mean	Median	Male (%)	Female (%)
CTRL	36	20–81	53.6	54	16 (44.4)	20 (55.6)
GBM	46	33–82	64.3	66	28 (60.9)	18 (39.1)
BM	28	42–82	63.5	62.6	18 (64.3)	10 (35.7)
M	28	30–79	58.6	60	5 (17.9)	23 (82.1)

## Data Availability

All datasets generated during the current study are available from the corresponding author upon reasonable request.

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
