# Peer review of "Raman Spectral Signatures of Serum-Derived Extracellular Vesicle-Enriched Isolates May Support the Diagnosis of CNS Tumors"

_cancers, 2021, doi:10.3390/cancers13061407_

Round 1
Reviewer 1 Report
In the manuscript entitled “Raman spectral signatures of serum-derived extracellular vesicles can support the diagnosis of CNS tumors” authors presented their work on the efficacy of Raman spectral signatures and multivariate statistical tools as discriminating parameters for central CNS malignancies such as glioblastoma multiforme, non-small cell lung cancer brain metastasis, meningioma and lumbar disc herniation as control. This is a well-planned and organized study and the manuscript is well structured too. Interestingly, the novelty of this exploratory study is employing extracellular vesicles derived from minimally invasive serum samples for classifying CNS malignances. However, I have following concerns:
- In isolation serum extracellular vehicles (EVs) the technique employed for isolation of the EVs is preliminary because literature shows that that the differential centrifugation technique could yield dismorphic EVs and also contaminants. Therefore, authors should present the data on purity and efficacy of the EVs (probably through TEM).
- Authors also should present the proofs for these EVs based on MISEV 2018 guidelines (atleeast 3 markers of EVs i.e. they should provide data on EV specific markers like CD81,CD63, Alix etc)
- Nanosight tracking analysis data should be produced to determine the size of the EV particles (60 -120 nm). If the particles are more than 150 nm they are not called EVs. So authors should show proof of this.
- Authors mentioned they took 5 spectra from each sample from 5 different sites. Raman spectra recording sites clarification is appreciated inorder to avoid duplicating.
- The optimal PC scores classification accuracy (Fig.3 ) are not matching with PC scores employed in the Fig.5. Authors should elaborate on this
- Authors should display the PC score plots that yielded best discrimination. For example PC2 Vs PC9 in CTRL Vs GBM. Similarly for others
- The main data is the subtraction spectra of Fig.6 needs more explanation inorder the exemplify the spectral differences in comparision with control. In addition a table of Raman spectral differences for each cancer type should be shown highlighting the major differences.
- Over all this study is very novel and well executed if the concerns are addressed it might improve the quality of MS further.
Note: I am curious to know whether it is possible to record Raman spectrum of single EV using optical tweezers and microraman spectroscopy ? If so can they present one spectrum of one EV for each sample.
Author Response
Reviewer #1
Dear Reviewer #1,
We highly appreciate your careful review, thoughtful suggestions on our manuscript. Here, we provide point-by-point answers (in italics) to each of your comments (in bold typeface). In the revised manuscript, all of the changes were done by the tracking function of the Microsoft Word to help the revision process.
Comments and Suggestions for Authors
In the manuscript entitled “Raman spectral signatures of serum-derived extracellular vesicles can support the diagnosis of CNS tumors” authors presented their work on the efficacy of Raman spectral signatures and multivariate statistical tools as discriminating parameters for central CNS malignancies such as glioblastoma multiforme, non-small cell lung cancer brain metastasis, meningioma and lumbar disc herniation as control. This is a well-planned and organized study and the manuscript is well structured too. Interestingly, the novelty of this exploratory study is employing extracellular vesicles derived from minimally invasive serum samples for classifying CNS malignances. However, I have the following concerns:
- In isolation serum extracellular vehicles (EVs) the technique employed for isolation of the EVs is preliminary because literature shows that that the differential centrifugation technique could yield dismorphic EVs and also contaminants. Therefore, authors should present the data on purity and efficacy of the EVs (probably through TEM).
- Authors also should present the proofs for these EVs based on MISEV 2018 guidelines (atleeast 3 markers of EVs i.e. they should provide data on EV specific markers like CD81, CD63, Alix etc)
- Nanosight tracking analysis data should be produced to determine the size of the EV particles (60 -120 nm). If the particles are more than 150 nm they are not called EVs. So, authors should show proof of this.
Addressing your first 3 concerns, we have added a further figure about the characterization of sEV samples (Figure 1) into the revised manuscript. This Figure includes the results of the NanoSight measurements, the Western blot analysis and the transmission electron microscopy. Based on the NanoSight results, our isolates match with the size range of small extracellular vesicles („[…]authors are urged to consider use of operational terms for EV subtypes that refer to physical characteristics of EVs, such as size (“small EVs” (sEVs) and “medium/large EVs” (m/lEVs), with ranges defined, for instance, respectively, < 100nm or < 200nm [small], or > 200nm [large and/or medium]))” – MISEV 2018)
As you mentioned, the methods employed can result in the co-isolation of contaminants. We would like to note that the aim of this study was to lay the basis for a cost-efficient, simple, and quick-to-implement procedure. Although some isolation methods, such as size exclusion chromatography and precipitation can be performed more quickly, isolation via differential centrifugation results in fewer particle in the size range potentially LPs, lower intensity of APOB and APOE LP markers, higher 61-150 nm EVs to 0-60 nm EVs ratio, and higher intensity of EV markers [75]. Although combinations of different methods would be the most appropriate approach, they could not be involved in a routine clinical practice due to their complexity, and result in an insufficient amount of the EVs from further analysis. Moreover, size exclusion chromatography and precision may involve higher costs.
Figure 1. Characterization of the sEV samples. The figure represents the results of the sEV characterization: size distribution of the sEV samples isolated from the four patient groups (black and red lines represent the mean and the standard deviation of the concentration, respectively) (A), a representative TEM image of the sEVs (B), and the Western Blot analysis of the sEV markers (C). (Abbreviations: CTRL, control; GBM, glioblastoma multiforme; BM, brain metastasis; M, meningioma; C, particle concentration; lys, cell lysate; Me, mean diameter size; Mo, mode diameter size.)
- Authors mentioned they took 5 spectra from each sample from 5 different sites. Raman spectra recording sites clarification is appreciated in order to avoid duplicating.
We are grateful that the Reviewer #1 has drawn our attention to this important issue. We have modified the description of the Raman measurements (rows 559-564).
Modified form in the manuscript:
“The spectra from all the samples were collected by using a 50× optical objective (Olympus). The described optical setup produces a laser spot of approx. 15 µm which is the sampling area of the Raman spectra and it is much smaller than the average size of the air-dried sample of approx. 4 mm, thus the Raman microscope operator can finely tune the position of the sampling spot and avoid duplication.”
- The optimal PC scores classification accuracy (Fig.3 ) are not matching with PC scores employed in the Fig.5. Authors should elaborate on this.
For Figure 5, to avoid overplotting, only the top 10 PCs were plotted, but the algorithm was performed with the number of PCs defined (rows 280-281). Otherwise, the position of the points would be difficult to see. We apologize if the description of the figure was not completely clear.
- Authors should display the PC score plots that yielded best discrimination. For example PC2 Vs PC9 in CTRL Vs GBM. Similarly for others
According to your suggestions, we have created four PCA score plots of the selected PCs (Figure S2). It should be noted that the selected PCs alone do not completely separate the groups, but are suitable for determining the main spectral differences.
Figure S2. PCA score plots of the selected PCs. This figure represents the score plots of the selected PCs in every comparison. (Abbreviations: CTRL, control; GBM, glioblastoma multiforme; BM, brain metastasis; M, meningioma; PC, principal component.)

Reviewer 2 Report
The manuscript entitled ‘Ramen Spectral Signatures of Serum Derived Extracellular Vesicles can Support the Diagnosis of CNS tumors’ is the first to report important data that supports the use of Raman spectroscopy analysis of blood derived extracellular vesicles in the diagnosis of CNS malignancies. Current methods used for the diagnosis of CNS malignancies include MRI imaging which cannot always diagnose tumor type (glioblastoma multiforme versus metastatic lesions) or prognosticate the disease, therefore invasive procedures such as biopsy are necessary to accurately diagnose the disease. There is a need for less invasive procedures to accurately diagnose and prognosticate malignancies.
Bukva et al. isolate small-extracellular vesicles (EVs) from the serum of 46 glioblastoma multiforme (GBM) patients, 28 brain metastasis (BM) of non-small cell lung carcinoma patients (NSCLC), 28 meningioma (M) patients and 36 lumbar disk herniation control (CTRL) patients by ultracentrifugation. Raman spectroscopy is then used to analyze the molecular composition of EVs from all groups. Raman spectroscopy found that the molecular composition of isolated EVs was unique enough to distinguish between not only CTRL patients and benign M and malignant disease GBM and BM patients, but also between BM patients and GBM patients with excellent specificity, sensitivity and AUC values. Furthermore, they identify the molecular components that the Raman spectral differences are likely attributed to. For example, spectral differences between CTRL vs GBM patients was attributed to carbohydrates and nucleic acids, CTRL vs BM EV spectral differences was due to lipids, and the spectral differences between CTRL vs M or BM vs GBM patients was attributed to acyl chains of lipids. The data overall show that Raman spectroscopy of EVs in patient plasma may be a useful way to diagnose CNS malignancies which are normally very difficult to diagnose without performing invasive procedures. The manuscript is excellently written and flows well.
Major:
- EVs are isolated by ultracentrifugation, the pellet is not washed and re-pelleted to remove co-isolated serum proteins and lipoproteins. The previously published manuscript indicated that serum proteins and lipoproteins are depleted in EVs as compared to serum but are present in the EV preparations (PMID 32731530). Addressing the concern about potential lipoprotein contamination of EV preparations would be useful for the discussion. Also, it would be helpful if the authors could comment on if there any studies out there that have used Raman Spectroscopy on lipoproteins – do their spectra differ significantly from EVs.
- Circulating nucleic acids can often be co-isolated with EVs and can even be of diagnostic value. As you observe that nucleic acids are potentially changing the Raman spectrum in GBM patient derived EVs as compared to CTRL EVs, have you treated the EVs with DNase and RNase to ensure that the Raman spectral differences are due to EVs and not co-isolated nucleic acids? It would be recommended to the authors to demonstrate if the spectrum is changed post-DNase/RNase treatment to address this.
- The NSCLC patients with brain metastasis have a unique spectra as compared to glioblastoma multiforme patients. If the authors can add a comment on whether the NSCLC patients were known to have any other residual disease (i.e. primary tumor/metastatic lesions) that would add addition support the spectral differences observed between these two patient groups are directly attribute to the brain metastatic lesions.
Minor:
- EVs are isolated by ultracentrifugation with reference back to a published manuscript (PMID 32731530) for EV characterization. Please include in your results section a small description of EV characterization in reference to size and concentration of EVs isolated from different patient populations as determined in the previous paper.
- Can you comment on the biological reasons why the EVs from these four patient groups may be molecularly distinct? i.e.: why might carbohydrate and nucleic acid EV content be different in EVs from control group as compared to GBM patients? Why might acyl lipids be important is distinguishing between EVs isolated from serum of BM and GBM patients?
- The author’s state in the conclusions (line 489) that the ultracentrifugation process followed by EV analysis only takes four hours and is therefore would be a promising protocol to use as a quick test in the clinic. The authors should consider the practicality of this and whether ultracentrifugation, even for 4hrs, would be likely to be implemented as a clinic-based testing platform. Especially as other techniques have been developed such as size exclusion chromatography and precipitation which would isolate EVs in a much shorter time period (30 minutes) and would potentially isolate a purer EV preparation. The authors should discuss this further.
Author Response
Reviewer #2
Dear Reviewer #2,
We highly appreciate your careful review, valuable comments and constructive feedback on our manuscript. Here, we provide point-by-point answers (in italics) to each of your comments (in bold typeface). In the revised manuscript, all of the changes were done by the tracking function of the Microsoft Word to help the revision process.
Comments and Suggestions for Authors
The manuscript entitled ‘Ramen Spectral Signatures of Serum Derived Extracellular Vesicles can Support the Diagnosis of CNS tumors’ is the first to report important data that supports the use of Raman spectroscopy analysis of blood derived extracellular vesicles in the diagnosis of CNS malignancies. Current methods used for the diagnosis of CNS malignancies include MRI imaging which cannot always diagnose tumor type (glioblastoma multiforme versus metastatic lesions) or prognosticate the disease, therefore invasive procedures such as biopsy are necessary to accurately diagnose the disease. There is a need for less invasive procedures to accurately diagnose and prognosticate malignancies.
Bukva et al. isolate small-extracellular vesicles (EVs) from the serum of 46 glioblastoma multiforme (GBM) patients, 28 brain metastasis (BM) of non-small cell lung carcinoma patients (NSCLC), 28 meningioma (M) patients and 36 lumbar disk herniation control (CTRL) patients by ultracentrifugation. Raman spectroscopy is then used to analyze the molecular composition of EVs from all groups. Raman spectroscopy found that the molecular composition of isolated EVs was unique enough to distinguish between not only CTRL patients and benign M and malignant disease GBM and BM patients, but also between BM patients and GBM patients with excellent specificity, sensitivity and AUC values. Furthermore, they identify the molecular components that the Raman spectral differences are likely attributed to. For example, spectral differences between CTRL vs GBM patients was attributed to carbohydrates and nucleic acids, CTRL vs BM EV spectral differences was due to lipids, and the spectral differences between CTRL vs M or BM vs GBM patients was attributed to acyl chains of lipids. The data overall show that Raman spectroscopy of EVs in patient plasma may be a useful way to diagnose CNS malignancies which are normally very difficult to diagnose without performing invasive procedures. The manuscript is excellently written and flows well.
Major:
- EVs are isolated by ultracentrifugation, the pellet is not washed and re-pelleted to remove co-isolated serum proteins and lipoproteins. The previously published manuscript indicated that serum proteins and lipoproteins are depleted in EVs as compared to serum but are present in the EV preparations (PMID 32731530). Addressing the concern about potential lipoprotein contamination of EV preparations would be useful for the discussion. Also, it would be helpful if the authors could comment on if there any studies out there that have used Raman Spectroscopy on lipoproteins – do their spectra differ significantly from EVs.
Based on your thoughtful suggestions, we modified the manuscript addressing the concern about potential lipoprotein contamination.
Modified form in the manuscript (rows 460-484):
It should be noted that co-purification of abundant serum proteins and lipoprotein (LP) particles in EV isolation methods is a common and well-known challenge. [71]. Liu and colleagues emphasized that serum is not the perfect choice for representative sampling of circulating EVs, as a high proportion of EVs may be lost during clotting, and blood components enrolled in the coagulation may also (e.g. platelets) can release EVs altering the original content of blood samples [58]. Some cancerous diseases, such as GBM, may also have a procoagulant phenotype [72].
Despite these difficulties, we have revealed in a previously published article that EV isolation from the serum samples of the same patient groups significantly the signal-to-noise ratio, even in the case of GBM with an elevated procoagulant activity [20]. Although abundant serum proteins and LPs were still present in EV isolates, isolation depleted their concentration and enriched the EV and tumor-specific protein markers. These results are consistent with previous similar researches on serum-derived EVs [19,73].
Enciso-Martinez and colleagues have determined Raman spectral signatures which were able to distinguish EVs from LPs and platelets with 95% confidence [74]. These special signature regions were found at 1004 cm-1 and between 2811 cm-1 and 3023 cm-1. Wavelength 1004 cm-1 had a strong peak in EVs but was not present in LPs and platelets. Furthermore, in the spectral range of 2811–3023 cm-1, EVs showed stronger intensity after 2900 cm-1 (‘protein component of the CH region’) compared to the spectra of LPs, where the region before 2900 cm-1 (’lipid component of the CH region’) proved to be more intense.
The Raman spectra of EVs isolated in our study show similar properties: a peak with strong intensity is present at 1004 cm-1 and the ’protein component’ of the CH region was found to be much more prominent than the ’lipid component’.
- Circulating nucleic acids can often be co-isolated with EVs and can even be of diagnostic value. As you observe that nucleic acids are potentially changing the Raman spectrum in GBM patient derived EVs as compared to CTRL EVs, have you treated the EVs with DNase and RNase to ensure that the Raman spectral differences are due to EVs and not co-isolated nucleic acids? It would be recommended to the authors to demonstrate if the spectrum is changed post-DNase/RNase treatment to address this.
Thank you for bringing a potentially misleading phenomenon to our attention.
As you mentioned, it is a known fact that RNA and DNA molecules can be co-isolated during vesicle isolation, as perhaps in our study. However, nucleic acids may be present in EV isolates not only due to co-isolation. Buzás et al. have shown that molecules associated with the surface of EVs include not only proteins and lipids, but also nuclease-sensitive nucleic acids, which should be considered as a structural component of vesicles [Buzás et al]. These nucleic acids may play an important role in tumor progression, as surface-associated DNA can alter the ability of EVs to adhere to fibronectin, suggesting that it may be helpful to determine how EVs interact with extracellular matrix molecules, such as those found in the tumor microenvironment or in the premetastatic niche [Jabalee et al.]. Taken together, might be that the Raman spectroscopy of sEVs detect DNA and RNA molecules attached to the surface of sEV, but those molecules are essential part of sEVs and describe the pathologic situation in the similar way than the protein and lipid components of sEVs.
Furthermore, treatment with RNase and DNase may introduce additional analytical noise into the Raman spectra due to the vibrations of the molecular bonds of the enzymes.
In view of your suggestions, treatment with RNase/DNase may be justified because of the potentially co-isolated nucleic acids, but thereby valuable tumor-specific information can be lost degrading the surface-associated DNA molecules while the analytical noise increases from the enzyme contamination.
Buzás et al: https://link.springer.com/article/10.1007/s00281-018-0682-0
Jabalee et al: https://www.ncbi.nlm.nih.gov/pmc/articles/PMC6115997/
- The NSCLC patients with brain metastasis have a unique spectra as compared to glioblastoma multiforme patients. If the authors can add a comment on whether the NSCLC patients were known to have any other residual disease (i.e. primary tumor/metastatic lesions) that would add addition support the spectral differences observed between these two patient groups are directly attribute to the brain metastatic lesions
In consultation with our neurosurgeon collaborator (Álmos Klekner), we can provide the following answer to your concern. This is a very important, thoughtful question if we discuss the experimental design and patient selection on a theoretical level. However, clinicians do not really encounter a patient with such a problem.
This is because metastasis can only originate from an existing primary tumor. Of course, it can’t happen that someone only has a brain metastasis, but there is no primary tumor in their body (since that’s where the tumor cells come from). All patients had lung tumors at the time of sampling. Either a postoperative residue or a newly diagnosed tumor. The volume of the primary tumor might be informative, but it would be an unfulfillable request: the volume of an amorphous and metastatic lung tumor cannot be precisely determined, it is not an acceptable task, but it has no significance.
In the light of the above, the Raman spectrum confirms the presence of an NSCLC tumor which resulted in a metastasis in the brain and could be very well separated from the GBM spectrum. Based on the experience and opinion of our neurosurgeon collaborators, the question would be significant and important in the case of NSCL and GBM common occurrence. However, there are almost no patients in the practice who suffer from both primary NSCLC tumor and GBM at the same time.
Minor:
- EVs are isolated by ultracentrifugation with reference back to a published manuscript (PMID 32731530) for EV characterization. Please include in your results section a small description of EV characterization in reference to size and concentration of EVs isolated from different patient populations as determined in the previous paper.
Please, find the Figure 1 about the characterization of sEV samples in the revised manuscript. This figure includes the results of the NanoSight measurements, the Western blot analysis and the transmission electron microscopy (rows 180-186).
Figure 1. Characterization of the sEV samples. The figure represents the results of the sEV characterization: size distribution of the sEV samples isolated from the four patient groups (black and red lines represent the mean and the standard deviation of the concentration, respectively) (A), a representative TEM image of the sEVs (B), and the Western Blot analysis of the sEV markers (C). (Abbreviations: CTRL, control; GBM, glioblastoma multiforme; BM, brain metastasis; M, meningioma; C, particle concentration; lys, cell lysate; Me, mean diameter size; Mo, mode diameter size.)
Can you comment on the biological reasons why the EVs from these four patient groups may be molecularly distinct? i.e.: why might carbohydrate and nucleic acid EV content be different in EVs from control group as compared to GBM patients? Why might acyl lipids be important is distinguishing between EVs isolated from serum of BM and GBM patients?
We have added further details to the revised manuscript to address this concern. Please find this in the rows 448-457:
Nevertheless, it was possible to identify discriminative spectral differences in the CTRL vs. GBM comparison, defining bands characteristic for carbohydrates and nucleic acids. These differences may be due to the characteristic metabolism of GBM, as it is associated with a significant increase in glycolysis for energy production and abnormal purine and pyrimidine synthesis [69]. Comparing the spectra of the CTRL and BM groups, significant differences were found in the characteristic bands of lipids and amino acids, which can be partly explained by the fact that an NSCLC appears to be reliant on fatty acid and serine catabolism [70]. Comparing the CTRL and M groups, as well as the two malignant groups BM and GBM, the lipid bands had outstanding importance with regard to discriminatory differences. The prominent importance of lipids in the BM vs. GBM comparison may be explained by the increased lipid catabolism of NSCLC and the elevated level of de novo lipid synthesis in GBM [69, 70]. However, more detailed identification based on the difficulties described above is not expedient for complex biological samples.
- The author’s state in the conclusions (line 489) that the ultracentrifugation process followed by EV analysis only takes four hours and is therefore would be a promising protocol to use as a quick test in the clinic. The authors should consider the practicality of this and whether ultracentrifugation, even for 4hrs, would be likely to be implemented as a clinic-based testing platform. Especially as other techniques have been developed such as size exclusion chromatography and precipitation which would isolate EVs in a much shorter time period (30 minutes) and would potentially isolate a purer EV preparation. The authors should discuss this further.
We appreciate this reasonable suggestion. You can find further added sentences in the revised manuscript that addresses your concern.
We would like to note that the aim of this study was to lay the basis for a cost-efficient, simple, and quick-to-implement procedure. Although some isolation methods, such as size exclusion chromatography and precipitation can be performed more quickly, isolation via differential centrifugation results in less particle in the size range of potentially LPs, and higher 61-150 nm EVs to 0-60 nm EVs ratio [75]. Although combinations of different methods would be the most appropriate approach, they could not be involved in a routine clinical practice due to their complexity, and result in an insufficient amount of the EVs from further analysis. Moreover, size exclusion chromatography and precipitation may involve higher costs. In terms of practicality, differential centrifugation is an appropriate compromise.
Modified form in the manuscript (rows 489-495):
Although some isolation methods, such as size exclusion chromatography and precipitation can be performed more quickly, isolation via differential centrifugation results in fewer particle in the size range potentially LPs, lower intensity of LP markers, higher 61-150 nm EVs to 0-60 nm EVs ratio, and higher intensity of EV markers. [75]. Although combinations of different methods would be the most appropriate approach, they could not be involved in a routine clinical practice due to their complexity and result in an insufficient amount of the EVs from further analysis. [75].

Reviewer 3 Report
I appreciate the efforts of the authors to setup EV liquid biopsy platform for CNS diseases such as GBM. However, it's clear that some fundamental characteristics of the disease were not taken into account during the design of the study. As the authors have mentioned in the introduction of their manuscript, brain tumors are particularly difficult to monitor. Some of the hurdles are biopsy is invasive, neuroimaging techniques are less accurate to monitor the progression of the disease, and importantly blood-brain barrier hinder the passage of tumor biomarkers from the brain to the circulation. So, I wonder if by collecting the blood, the authors would have sufficient tumor biomarkers to analyze and to increase the signal-to-noise ratio during the measurement (Figure 6).
Another characteristic of GBM is that this disease is associated with a highly procoagulant phenotype, and in some cases with high levels of tissue factor, the main protein which initiates coagulation cascade. Some EVs have been shown to have clotting properties (see Risada & Mackman Res Pract Thromb Haemost. 2019, Gardiner et al JEV2015). During serum preparation, blood was clotted. This means that some EVs may have been lost during preparation. Thus, the results authors presented may not fully reflect the disease characters.
Furthermore, I also have a concern about the isolation procedure of sEVs use in this study. By differential centrifugations, non-EV contaminants such as RNA and other proteins may get sedimented disturbing the measurement of EVs. Authors also said that they measured EVs using AFM, but I cannot find the result of this measurement in their manuscript.
Thus, although the idea of this study was promising, but the design of this study (sample collection and preparation) in my opinion can be much improved.
Author Response
Reviewer #3
Dear Reviewer #3,
We are grateful to Reviewer #3 for his thoughtful comments and remarks. Here, we provide point-by-point answers (in italics) to each of your comments (in bold typeface). In the revised manuscript, all of the changes were done by the tracking function of the Microsoft Word to help the revision process.
Comments and Suggestions for Authors
I appreciate the efforts of the authors to setup EV liquid biopsy platform for CNS diseases such as GBM. However, it's clear that some fundamental characteristics of the disease were not taken into account during the design of the study. As the authors have mentioned in the introduction of their manuscript, brain tumors are particularly difficult to monitor. Some of the hurdles are biopsy is invasive, neuroimaging techniques are less accurate to monitor the progression of the disease, and importantly blood-brain barrier hinder the passage of tumor biomarkers from the brain to the circulation. So, I wonder if by collecting the blood, the authors would have sufficient tumor biomarkers to analyze and to increase the signal-to-noise ratio during the measurement (Figure 6).
Another characteristic of GBM is that this disease is associated with a highly procoagulant phenotype, and in some cases with high levels of tissue factor, the main protein which initiates coagulation cascade. Some EVs have been shown to have clotting properties (see Risada & Mackman Res Pract Thromb Haemost. 2019, Gardiner et al JEV2015). During serum preparation, blood was clotted. This means that some EVs may have been lost during preparation. Thus, the results authors presented may not fully reflect the disease characters.
As you wrote, identifying blood-based CNS tumor markers is challenging, presumably explained by several reasons. One of the most limiting factor is the blood-brain-barrier (BBB), which can prevent tumor-derived molecules from entering the peripheral blood. However, despite these difficulties, EVs are a promising source of CNS tumor biomarkers during a liquid biopsy. This is possible because EVs (including small EVs) are presumably able to cross the BBB through transcytosis, carrying number of tumor-specific molecules (DOI: 10.1227/NEU.0000000000001242; DOI: 10.18632/oncotarget.13635).
This approach can also be applied to GBM. Research has found that several GBM-specific biomarkers carry extracellular vesicles in circulation (such as glioma-specific mutant EGFR variant III, metylathed promoter region of O-6-methyl-guanine-DNA methyltransferase or 24S-hydroxycholesterol) (DOI: 10.1227/NEU.0000000000001242; DOI: 10.18632/oncotarget.13635).
Along the line of these results, in our previously published article we have demonstrated that the EV isolation significantly improve the signal-to-noise ratio comparing the serum and the EV samples of the same patient groups. Although abundant serum proteins and lipoproteins were still present in EV isolates, isolation depleted their concentration and enriched the EV-markers and tumor-specific protein markers. This was reflected in both the clustering property of the samples and the in silico function prediction (DOI: 10.3390/ijms21155359). In the same study, we also examined the presence of characteristic protein biomarkers. Based on the results, sEVs were successfully isolated from samples from all patient groups (even from GBM samples, which tends to show high procagulant activity.)
In light of these findings, it seems warranted to examine circulating sEV in order to develop liquid biopsy-based diagnostic methods for CNS tumors.
Furthermore, I also have a concern about the isolation procedure of sEVs use in this study. By differential centrifugations, non-EV contaminants such as RNA and other proteins may get sedimented disturbing the measurement of EVs. Authors also said that they measured EVs using AFM, but I cannot find the result of this measurement in their manuscript.
As you mentioned, the methods employed can result in the co-isolation of contaminants. We would like to note that the aim of this study was to lay the basis for a cost-efficient, simple, and quick-to-implement procedure. Although some isolation methods, such as size exclusion chromatography and precipitation can be performed more quickly, isolation via differential centrifugation results in fewer particle in the size range potentially LPs, lower intensity of APOB and APOE LP markers, higher 61-150 nm EVs to 0-60 nm EVs ratio, and higher intensity of EV markers [75].
It may well be that size exclusion chromatography and precipitation result in lower co-isolated protein concentration, but do not involve higher amounts of EV particles, meanwhile the number of LP particles is significantly higher compared to differential centrifugation [75].
Although combinations of different methods would be the most appropriate approach, they could not be involved in a routine clinical practice due to their complexity, and result in an insufficient amount of the EVs from further analysis. Moreover, size exclusion chromatography and precision may involve higher costs. In terms of practicality, differential centrifugation is an appropriate compromise.
As other reviewers also expressed concerns about the characterization and isolation of sEV samples, we also performed the EV characterization in this study. Please, find the Figure 1, rows 180-186 in the Results:
Isolated sEVs were characterized by transmission electron microscopy (TEM) and nanoparticle tracking analysis (NTA), as well as by examining characteristic sEV markers (Alix, CD81 and calnexin) by Western blotting (WB) (Figure 1). Average concentration, mean and mode diameter of the particles were measured as 7.41×1010 particles/ml, 111.20 nm and 83.32 nm, respectively. Alix, CD81 positivity and calnexin negativity was determined (see the Figure S1 for the original WB images).
Figure 1. Characterization of the sEV samples. The figure represents the results of the sEV characterization: size distribution of the sEV samples isolated from the four patient groups (black and red lines represent the mean and the standard deviation of the concentration, respectively) (A), a representative TEM image of the sEVs (B), and the Western Blot analysis of the sEV markers (C). (Abbreviations: CTRL, control; GBM, glioblastoma multiforme; BM, brain metastasis; M, meningioma; C, particle concentration; lys, cell lysate; Me, mean diameter size; Mo, mode diameter size.)
Based on the results, the EV isolation resulted in particles with mean diameter of 111.20 nm and mode diameter of 83.32 nm. The characteristic of the Raman spectra for the EV samples also suggest the predominance of EV amount over LP concentration (rows 489-495):
Enciso-Martinez and colleagues have determined Raman spectral signatures which were able to distinguish EVs from LPs and platelets with 95% confidence [74]. These special signature regions were found at 1004 cm-1 and between 2811 cm-1 and 3023 cm-1. Wavelength 1004 cm-1 had a strong peak in EVs but was not present in LPs and platelets. Furthermore, in the spectral range of 2811–3023 cm-1, EVs showed stronger intensity after 2900 cm-1 (‘protein component of the CH region’) compared to the spectra of LPs, where the region before 2900 cm-1 (’lipid component of the CH region') proved to be more intense.
The Raman spectra of EVs isolated in our study show similar properties: a peak with strong intensity is present at 1004 cm-1 and the ’protein component’ of the CH region was found to be much more prominent than the ’lipid component’.

Reviewer 4 Report
The authors provided a well-documented work providing an interesting starting point for exosome characterization. In this regard, I suggest add some words about the need to design new approaches for exosome diversity characterization focused on the understanding of exosome origin and function ( PMID: 32759810 and PMID: 29029605 are just as an example).
I look forward to receiving the updated version of the paper.
Author Response
Reviewer #4
Comments and Suggestions for Authors
The authors provided a well-documented work providing an interesting starting point for exosome characterization. In this regard, I suggest add some words about the need to design new approaches for exosome diversity characterization focused on the understanding of exosome origin and function ( PMID: 32759810 and PMID: 29029605 are just as an example).I look forward to receiving the updated version of the paper.
Dear Reviewer #4,
We highly appreciate your constructive feedback on our manuscript. Based on your suggestions, we have added further sentences in the revised manuscript (row 143-146):
In addition, Raman spectroscopy may be suitable for the characterization of EVs by identifying different subtypes by origin and function, which is an important and long-standing challenge for EV-based biomarker research [44].

Round 2
Reviewer 2 Report
Reviewer comments/concerns have been sufficiently addressed by the authors.
Author Response
Thank you for your kindly, positive review.
Reviewer 3 Report
The revised version of the manuscript is still not satisfying as the authors didn't really elaborate the reasons with supporting experiments why serum is superior than plasma as the source of EV biomarkers in their GBM patient population (those with procoagulant phenotypes).
They showed a representative TEM image in Figure 1B. In my opinion, this TEM image has a really poor resolution. In their method section (548-552), it was not clear how they prepared EV on the grid and how they stained their EV specimen. EVs imaged with TEM at dry stage would commonly give a cup shape appearance as they would collapse during drying and staining (Yuana et al JEV 2015). Thus, authors couldn't exclude that these particles in Figure 1B are actually non-EVs. I strongly advice that the authors check their EV isolates for non-EV contaminants e.g. lipoproteins or to use size exclusion chromatography to compare results and confirm their findings.
Though the measurement using Raman spectroscopy is promising, authors have to think carefully about the preparation of EV samples which is actually much more critical than just the measurement of samples itself. The use of ultracentrifugation for EV isolation is not suitable for a quick test in the clinic as small laboratories are likely not equipped with ultracentrifugation. Furthermore, isolated EVs were re-suspended in DPBS and frozen/stored in -80C. Wouldn't it be more cost-efficient and less artefactual contamination if EVs could be measured fresh? Why didn't the author freeze the serum samples instead and performed the EV isolation just before the measurement? All of these considerations should be mentioned in their manuscript.
Author Response
Reviewer #3
Dear Reviewer #3,
We are grateful for your thoughtful comments and remarks. Here, we provide answers (in italic) to your comments (in bold). In the revised manuscript, all of the changes were done by the tracking function of Microsoft Word to help the revision process.
Comments and Suggestions for Authors
The revised version of the manuscript is still not satisfying as the authors didn't really elaborate the reasons with supporting experiments why serum is superior than plasma as the source of EV biomarkers in their GBM patient population (those with procoagulant phenotypes).
They showed a representative TEM image in Figure 1B. In my opinion, this TEM image has a really poor resolution. In their method section (548-552), it was not clear how they prepared EV on the grid and how they stained their EV specimen. EVs imaged with TEM at dry stage would commonly give a cup shape appearance as they would collapse during drying and staining (Yuana et al JEV 2015). Thus, the authors couldn't exclude that these particles in Figure 1B are actually non-EVs. I strongly advice that the authors check their EV isolates for non-EV contaminants e.g. lipoproteins or to use size exclusion chromatography to compare results and confirm their findings.
According to your concerns, we modified the description of the TEM (rows 591-593):
“For TEM measurements the samples were dropped on a grid (carbon film with 200 Mesh copper grids (CF200-Cu, Electron Microscopy Sciences, USA) and dried without staining or other fixation procedure.”
As you rightly noted, cup-shaped EVs can occur in TEM images. Rikkert et al. have shown that different TEM protocols yield different proportions of cup-shaped EVs (3-96%), and protocols without surface fixation resulted in the lowest proportion of them (DOI: 10.1080/20013078.2018.1555419, JEV 2019). These results are consistent with the findings described in another study that the cup-shaped EV artifact can be caused by surface fixation (and dehydration) (DOI 10.1007/s00216-015-8535-3). In the light of these, the lack of cup-shaped EVs could be because no surface fixation procedure was applied in the TEM measurements.
Reviewer #3 has real and thoughtful concerns about the presence of LPs. However, confirmation by size exclusion chromatography is not recommended according to the published literature. Working with serum samples, Brennan et al. have compared several isolation protocols, including ultracentrifugation and two size exclusion chromatography kits [75] (JEV 2020, DOI: 10.1038/s41598-020-57497-7). Based on their results, the concentration of LP particles was significantly higher using the size exclusion chromatography than the ultracentrifugation, which was also confirmed by examining LP markers (APOE, APOB). In addition, the intensities of CD63 and TSG101 EV markers were significantly lower in size exclusion chromatography isolates than in the ultracentrifugation isolates. Based on their results, they concluded that size exclusion chromatography co-isolated more LP particles from serum than ultracentrifugation.
In our previously proteomics-based article, we have shown that EV isolation from the serum of the sampe patient groups via differential centrifugation reduced significantly (with fold change of 0.5-2) the apolipoprotein concentration, and enriched the non-tissue specific EV marker proteins (with fold change of 0.5-8) [20] (Dobra et al., 2020, IJMS). However, it should be noted that the LP fraction could not be completely eliminated. (This phenomenon is mentioned in our manuscript in the row 492.)
Though the measurement using Raman spectroscopy is promising, authors have to think carefully about the preparation of EV samples which is actually much more critical than just the measurement of samples itself. The use of ultracentrifugation for EV isolation is not suitable for a quick test in the clinic as small laboratories are likely not equipped with ultracentrifugation. Furthermore, isolated EVs were re-suspended in DPBS and frozen/stored in -80C. Wouldn't it be more cost-efficient and less artefactual contamination if EVs could be measured fresh? Why didn't the author freeze the serum samples instead and performed the EV isolation just before the measurement? All of these considerations should be mentioned in their manuscript.
There were only technical reasons for storing the samples at -80 °C, as serum samples from 138 patients arrived from the Neurosurgery Clinic of Debrecen at the same time. It was not in our power to isolate EVs from such a large amount of serum samples and simultaneously perform the Raman spectroscopic measurements. Using filtered (0.22 um) DPSB for resuspension is a general recommendation for examining small extracellular vesicles through nanoparticle tracking analysis in the size range of 50-120 nm (Parsons et al. 2017; DOI: 10.3389/fcvm.2017.00068).
All things considered, we fully understand Reviewer #3’s concerns about isolation, contaminants, and clinical applicability. In line with these concerns, we attempted to avoid any statement describing the current method as clinically applicable in our manuscript. In the revised version, we highlighted and emphasized that the method needs further development in the future and may only provide a theoretical basis for a future diagnostic tool.
Accordingly, rows 66-68 have been modified as follows:
“Our results support that Raman spectroscopic analysis of sEVs is a promising method that could be further developed in order to be applicable in the diagnosis of CNS tumors.”
We previously highlighted that LPs and abundant serum proteins are still present in the isolates (row 492). Now, we drew attention to the fact that examining plasma samples is more expedient in the future, and cited Liu et al. [58] (2019, Blood Plasma versus Serum: Which Is Right for Sampling Circulating Membrane Microvesicles in Human Subjects?) and Smolarz et al [71] (2011, Proteome Profiling of Exosomes Purified from a Small Amount of Human Serum: The Problem of Co-Purified Serum Components) In the rows 494-496:
[…] examining plasma instead of serum should be considered in further investigations [58, 71].
In rows 533-535, we also emphasized that our results can only provide a theoretical basis for a future method, but further improvements are needed:
“In conclusion, our results provide a proof of principle for a novel detection technology that might be utilized to develop a relatively easy-to-execute and appropriate method, which could have the potential to support and simplify the diagnosis and monitoring of CNS tumors in the future.”
As well as we modified the Conclusion (rows 674-676):
“In conclusion, our results support that Raman spectroscopic analysis of circulating sEVs is a promising liquid-biopsy based method that could be further developed in order to be applicable in the diagnosis of CNS tumors.”

Reviewer 4 Report
The article is now ready for publication.
Author Response

(The authors gave the same response as above.)
